# Why Normalizing Flows Fail to Detect Out-of-Distribution Data

**Polina Kirichenko**$^*$
pk1822@nyu.edu
New York University

**Pavel Izmailov**$^*$
pi390@nyu.edu
New York University

**Andrew Gordon Wilson**
andrewgw@cims.nyu.edu
New York University

## Abstract

Detecting out-of-distribution (OOD) data is crucial for robust machine learning systems. Normalizing flows are flexible deep generative models that often surprisingly fail to distinguish between in- and out-of-distribution data: a flow trained on pictures of clothing assigns higher likelihood to handwritten digits. We investigate why normalizing flows perform poorly for OOD detection. We demonstrate that flows learn local pixel correlations and generic image-to-latent-space transformations which are not specific to the target image datasets, focusing on flows based on coupling layers. We show that by modifying the architecture of flow coupling layers we can bias the flow towards learning the semantic structure of the target data, improving OOD detection. Our investigation reveals that properties that enable flows to generate high-fidelity images can have a detrimental effect on OOD detection.

## 1 Introduction

Normalizing flows [42, 9, 10] seem to be ideal candidates for out-of-distribution detection, since they are simple generative models that provide an exact likelihood. However, Nalisnick et al. [29] revealed the puzzling result that flows often assign higher likelihood to out-of-distribution data than the data used for maximum likelihood training. In Figure 1(a), we show the log-likelihood histogram for a RealNVP flow model [10] trained on the ImageNet dataset [37] subsampled to $64 \times 64$ resolution. The flow assigns higher likelihood to both the CelebA dataset of celebrity photos, and the SVHN dataset of images of house numbers, compared to the target ImageNet dataset.

While there has been empirical progress in improving OOD detection with flows [29, 7, 30, 39, 40, 48], the fundamental reasons for why flows fail at OOD detection in the first place are not fully understood. In this paper, we show how the *inductive biases* [28, 47] of flow models — implicit assumptions in the architectures and training procedures — can hinder OOD detection.

In particular, our contributions are the following:

- We show that flows learn latent representations for images largely based on local pixel correlations, rather than semantic content, making it difficult to detect data with anomalous semantics.
- We identify mechanisms through which normalizing flows can simultaneously increase likelihood for all structured images. For example, in Figure 1(b, c), we show that the coupling layers of RealNVP transform the in-distribution ImageNet in the same way as the OOD CelebA.
- We show that by changing the architectural details of the coupling layers, we can encourage flows to learn transformations specific to the target data, improving OOD detection.

---

$^*$Equal contribution

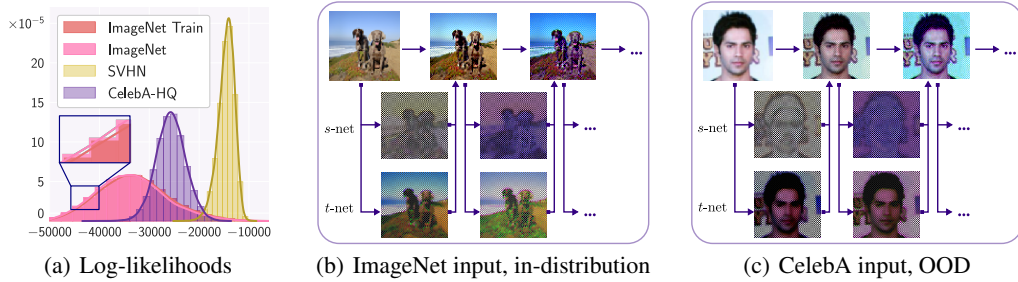

| (a) Log-likelihoods | (b) ImageNet input, in-distribution | (c) CelebA input, OOD |

Figure 1: **RealNVP flow on in- and out-of-distribution images.** (**a**): A histogram of log-likelihoods that a RealNVP flow trained on ImageNet assigns to ImageNet, SVHN and CelebA. The flow assigns higher likelihood to out-of-distribution data. (**b**, **c**): A visualization of the intermediate layers of a RealNVP model on an (b) in-distribution image and (c) OOD image. The first row shows the coupling layer activations, the second and third rows show the scale $s$ and shift $t$ parameters predicted by a neural network applied to the corresponding coupling layer input. Both on in-distribution and out-of-distribution images, $s$ and $t$ accurately approximate the structure of the input, even though the model has not observed inputs (images) similar to the OOD image during training. *Flows learn generic image-to-latent-space transformations that leverage local pixel correlations and graphical details rather than the semantic content needed for OOD detection.*

- We show that OOD detection is improved when flows are trained on high-level features which contain semantic information extracted from image datasets.

We also provide code at `https://github.com/PolinaKirichenko/flows_ood`.

## 2 Background

We briefly introduce normalizing flows based on coupling layers. For a more detailed introduction, see Papamakarios et al. [33] and Kobyzev et al. [24].

**Normalizing flows** Normalizing flows [42] are a flexible class of deep generative models that model a target distribution $p^*(x)$ as an invertible transformation $f$ of a base distribution $p_Z(z)$ in the latent space. Using the change of variables formula, the likelihoods for an input $x$ and a dataset $\mathcal{D}$ are

$$p_X(x) = p_Z(f^{-1}(x)) \left| \det \frac{\partial f^{-1}}{\partial x} \right|, \quad p(\mathcal{D}) = \prod_{x \in \mathcal{D}} p_X(x). \tag{1}$$

The latent space distribution $p_Z(z)$ is commonly chosen to be a standard Gaussian. Flows are typically trained by maximizing the log-likelihood (1) of the training data with respect to the parameters of the invertible transformation $f$.

**Coupling layers** We focus on normalizing flows based on *affine coupling layers*. In these flows, the transformation performed by each layer is given by

$$f_{\text{aff}}^{-1}(x_{\text{id}}, x_{\text{change}}) = (y_{\text{id}}, y_{\text{change}}), \quad \begin{cases} y_{\text{id}} = x_{\text{id}} \\ y_{\text{change}} = (x_{\text{change}} + t(x_{\text{id}})) \odot \exp(s(x_{\text{id}})) \end{cases} \tag{2}$$

where $x_{\text{id}}$ and $x_{\text{change}}$ are disjoint parts of the input $x$, $y_{\text{id}}$ and $y_{\text{change}}$ are disjoint parts of the output $y$, and the scale and shift parameters $s(\cdot)$ and $t(\cdot)$ are usually implemented by a neural network (which we will call the *st-network*). The split of the input into $x_{\text{id}}$ and $x_{\text{change}}$ is defined by a *mask*: a coupling layer transforms the masked part $x_{\text{change}} = \text{mask}(x)$ of the input based on the remaining part $x_{\text{id}}$. The transformation (2) is invertible and allows for efficient Jacobian computation in (1):

$$\log \left| \det \frac{\partial f_{\text{aff}}^{-1}}{\partial x} \right| = \sum_{i=1}^{\dim(x_{\text{change}})} s(x_{\text{id}})_i. \tag{3}$$

**Flows with coupling layers** Coupling layers can be stacked together into flexible normalizing flows: $f = f^K \circ f^{K-1} \circ \ldots \circ f^1$. Examples of flows with coupling layers include NICE [9], RealNVP [10], Glow [23], and many others [e.g., 4, 5, 11, 16, 17, 22, 26, 34].

**Out-of-distribution detection using likelihood**    Flows can be used for out-of-distribution detection based on the likelihood they assign to the inputs. One approach is to choose a likelihood threshold $\epsilon$ on a validation dataset, e.g. to satisfy a desired false positive rate, and during test time identify inputs which have likelihood lower than $\epsilon$ as OOD. Qualitatively, we can estimate the performance of the flows for OOD detection by plotting a histogram of the log-likelihoods such as Figure 1(a): the likelihoods for in-distribution data should generally be higher compared to OOD. Alternatively, we can treat OOD detection as a binary classification problem using likelihood scores, and compute accuracy with a fixed likelihood threshold $\epsilon$, or AUROC (area under the receiver operating characteristic curve).

## 3    Related Work

Recent works have shown that normalizing flows, among other deep generative models, can assign higher likelihood to out-of-distribution data [29, 7]. The work on OOD detection with deep generative models falls into two distinct categories. In group anomaly detection (GAD), the task is to label a batch of $n > 1$ datapoints as in- or out-of-distribution. Point anomaly detection (PAD) involves the more challenging task of labelling single points as out-of-distribution.

**Group anomaly detection**    Nalisnick et al. [30] introduce the typicality test which distinguishes between a high density set and a typical set of a distribution induced by a model. However, the typicality test cannot detect OOD data if the flow assigns it with a similar likelihood distribution to that of in-distribution data. Song et al. [40] showed that out-of-distribution datasets have lower likelihoods when batch normalization statistics are computed from a current batch instead of accumulated over the train set, and proposed a test based on this observation. Zhang et al. [48] introduce a GAD algorithm based on measuring correlations of flow's latent representations corresponding to the input batch. The main limitation of GAD methods is that for most practical applications the assumption that the data comes in batches of inputs that are all in-distribution or all OOD is not realistic.

**Point anomaly detection**    Choi et al. [7] proposed to estimate the Watanabe-Akaike Information Criterion using an ensemble of generative models, showing accurate OOD detection on some of the challenging dataset pairs. Ren et al. [35] explain the poor OOD detection performance of deep generative models by the fact that the likelihood is dominated by background statistics. They propose a test based on the ratio of the likelihoods for the image and background likelihood estimated using a separate *background model*. Serrà et al. [39] show that normalizing flows assign higher likelihoods to simpler datasets and propose to normalize the flow's likelihood by an image complexity score.

In concurrent work, Schirrmeister et al. [38] find that invertible flows learn low-level features which dominate the likelihood which is consistent with our results.

In this work we argue that it is the inductive biases of the model that determine its OOD performance. While most work treats flows as black-box density estimators, we conduct a careful study of the latent representations and image-to-latent-space transformations learned by the flows. Throughout the paper, we connect our findings with prior work and provide new insights.

## 4    Why flows fail to detect OOD data

Normalizing flows consistently fail at out-of-distribution detection when applied to common benchmark datasets (see Appendix D). In this paper, we discuss the reasons behind this surprising phenomenon. We summarize our thesis as follows:

> The maximum likelihood objective has a limited influence on OOD detection, relative to the *inductive biases* of the flow, captured by the modelling assumptions of the architecture.

**Why should flows be able to detect OOD inputs?**    Flows are trained to maximize the likelihood of the training data. Likelihood is a probability density function $p(\mathcal{D})$ defined on the image space and hence has to be normalized. Thus, likelihood cannot be simultaneously increased for all the inputs (images). In fact, the optimal maximizer of (1) would only assign positive density to the datapoints in the training set, and, in particular, would not even generalize to the test set of the same dataset. In practice, flows do not seem to overfit, assigning similar likelihood distributions to train and and test

(see e.g. Figure 1(a)). Thus, despite their flexibility, flows are not maximizing the likelihood (1) to values close to the global optimum.

**High density and typical sets** In prior work, poor performance of normalizing flows in out-of-distribution detection is explained by the discrepancy between *typical* and *high-density sets* of a generative model [30]. For example, samples from a standard Gaussian distribution $\mathcal{N}(0, I)$ in high dimensions are concentrated in a thin spherical shell of radius $\sqrt{d}$ where $d$ is the dimension of the space, and high-density points near the origin do not belong to the typical set. This phenomenon explains how generative models can produce sound samples and at the same time assign the highest likelihood to atypical data. However, the discrepancy between the high density and typical sets is a very general phenomenon and the specific form that it takes in complex deep generative models such as normalizing flows is not clear. In particular, the typicality arguments do not provide insights into how normalizing flows distribute the likelihood and specifically why datasets of structured natural images receive high likelihood.

**What is OOD data?** There are infinitely many distributions that give rise to any value of the likelihood objective in (1) except the global optimum. Indeed, any non-optimal solution assigns probability mass outside of the training data distribution; we can arbitrarily re-assign this probability mass to get a new solution with the same value of the objective (see Appendix A for a detailed discussion). Therefore the inductive biases of a model determines which specific solution is found through training. In particular, the inductive biases will affect what data is assigned high likelihood (in-distribution) and what data is not (OOD).

**What inductive biases are needed for OOD detection?** The datasets in computer vision are typically defined by the semantic content of the images. For example, the CelebA dataset consists of images of faces, and SVHN contains images of house numbers. In order to detect OOD data, the inductive biases of the model have to be aligned with learning the semantic structure of the data, i.e. what objects are represented in the data.

**What are the inductive biases of normalizing flows?** In the remainder of the paper, we explore the inductive biases of normalizing flows. We argue that flows are biased towards learning *graphical* properties of the data such as local pixel correlations (e.g. nearby pixels usually have similar colors) rather than semantic properties of the data (e.g. what objects are shown in the image).

**Flows have capacity to distinguish datasets** In Appendix B, we show that if we explicitly train flows to distinguish between a pair of datasets, they can assign large likelihood to one dataset and low likelihood to the other. However, when trained with the standard maximum likelihood objective, flows do not learn to make this distinction. The inductive biases of the flows prefer solutions that assign high likelihood to most structured datasets simultaneously.

## 5 Flow latent spaces

Normalizing flows learn highly non-linear image-to-latent-space mappings often using hundreds of millions of parameters. One could imagine that the learned latent representations have a complex structure, encoding high-level semantic information about the inputs. In this section, we visualize the learned latent representations on both in-distribution and out-of-distribution data and demonstrate that they encode simple graphical structure rather than semantic information.

> **Observation**: There exists a correspondence between the coordinates in an image and in its learned representation. We can recognize edges of the inputs in their latent representations.
> **Significance for OOD detection:** In order to detect OOD images, a model has to assign likelihood based on the semantic content of the image (see Sec. 4). Flows do not represent images based on their semantic contents, but rather directly encode their visual appearance.

In the first four columns of Figure 2, we show latent representations[2] of a RealNVP model trained on FashionMNIST for an in-distribution FashionMNIST image and an out-of-distribution MNIST digit. The first column shows the original image $x$, and the second column shows the corresponding latent $z$. The latent representations appear noisy both for in- and out-of-distribution samples, but the edges

of the MNIST digit can be recognized in the latent. In the third column of Figure 2, we show latent representations averaged over $K = 40$ samples of dequantization noise[3] $\epsilon_k$: $\frac{1}{K}\sum_{k=1}^{K} f^{-1}(x + \epsilon_k)$. In the averaged representation, we can clearly see the edges from the original image. Finally, in the fourth column of Figure 2, we visualize the latent representations (for a single sample of dequantization noise) from a flow when batch normalization layers are in train mode [19]. In train mode, batch normalization layers use the activation statistics of the current batch, and in evaluation mode they use the statistics accumulated over the train set. While for in-distribution data there is no structure visible in the latent representation, the out-of-distribution latent clearly preserves the shape of the 7-digit from the input image. In the remaining panels of Figure 2, we show an analogous visualization for a RealNVP trained on CelebA using an SVHN image as OOD. In the third panel of this group, we visualize the blue channel of the latent representations. Again, the OOD input can be recognized in the latent representation; some of the edges from the in-distribution CelebA image can also be seen in the corresponding latent variable. Additional visualizations (e.g. for Glow) are in Appendix F.

**Insights into prior work**     The group anomaly detection algorithm proposed in Zhang et al. [48] uses correlations of the latent representations as an OOD score. Song et al. [40] showed that normalizing flows with batch normalization layers in train mode assign much lower likelihood to out-of-distribution images than they do in evaluation mode, while for in-distribution data the difference is not significant. Our visualizations explain the presence of correlations in the latent space and shed light into the difference between the behaviour of the flows in train and test mode.

## 6    Transformations learned by coupling layers

To better understand the inductive biases of coupling-layer based flows, we study the transformations learned by individual coupling layers.

**What are coupling layers trained to do?**     Each coupling layer updates the masked part $x_{\text{change}}$ of the input $x$ to be $x_{\text{change}} \leftarrow (x_{\text{change}} + t(x_{\text{id}})) \cdot \exp(s(x_{\text{id}}))$, where $x_{\text{id}}$ is the non-masked part of $x$, and $s$ and $t$ are the outputs of the $st$-network given $x_{\text{id}}$ (see Section 2). The flow is encouraged to predict high values for $s$ since for a given coupling layer the Jacobian term in the likelihood of Eq. (1) is given by $\sum_j s(x_{\text{id}})_j$ (see Section 4). Intuitively, to afford large values for scale $s$ without making the latent representations large in norm and hence decreasing the density term $p_{\mathcal{Z}}(z)$ in (1), the shift $-t$ has to be an accurate approximation of the masked input $x_{\text{change}}$. For example, in Figure 1(b, c) the $-t$ outputs of the first coupling layers are a very close estimate of the input to the coupling layer. The likelihood for a given image will be high whenever the coupling layers can accurately predict masked pixels. To the best of our knowledge, this intuition has not been discussed in any previous work.

> **Observation**: We describe two mechanisms through which coupling layers learn to predict the masked pixels: (1) leveraging local color correlations and (2) using information about the masked pixels encoded by the previous coupling layer (coupling layer co-adaptation).
> **Significance for OOD detection**: These mechanisms allow the flows to predict the masked pixels equally accurately on in- and out-of-distribution datasets. As a result, flows assign high likelihood to OOD data.

### 6.1    Leveraging local pixel correlations

In Figure 3(a, b), we visualize intermediate coupling layer activations of a small RealNVP model with 2 coupling layers and checkerboard masks trained on FashionMNIST. For the masked inputs, the outputs of the $st$-network are shown in black. Even though the flow was trained on FashionMNIST and has never seen an MNIST digit, the $st$-networks can easily predict masked from observed pixels on both FashionMNIST and MNIST. Figure 1 shows the same behaviour in the first coupling layers of RealNVP trained on ImageNet.

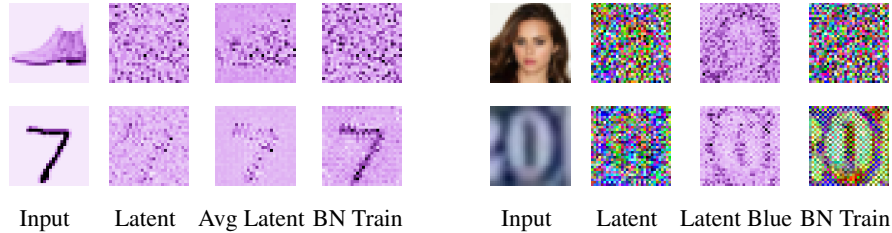

Input  Latent  Avg Latent  BN Train          Input  Latent  Latent Blue  BN Train

Figure 2: **Latent spaces.** Visualization of latent representations for the RealNVP model on in-distribution and out-of-distribution inputs. Panels 1-4: original images, latent representations, latent representation averaged over 40 samples of dequantization noise, and latent representations for batch normalization in train mode for a flow trained on FashionMNIST and using MNIST for OOD data. Panels 5-8: same as 1-4 but for a model trained on CelebA with SVHN for OOD, except in panel 7 we show the blue channel of the latent representation from panel 6 instead of an averaged latent representation. For both dataset pairs, we can recognize the shape of the input image in the latent representations. The flow represents images based on their graphical appearance rather than semantic content.

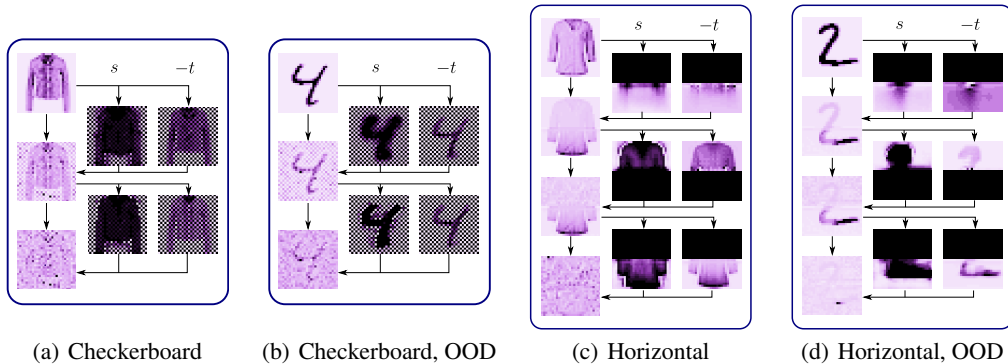

(a) Checkerboard  (b) Checkerboard, OOD  (c) Horizontal  (d) Horizontal, OOD

Figure 3: **Coupling layers.** Visualization of RealNVP's intermediate coupling layer activations, as well as scales $s$ and shifts $t$ predicted by each coupling layer on in-distribution (panels a, c) and out-of-distribution inputs (panels b, d). RealNVP was trained on FashionMNIST. (a), (b): RealNVP with a standard checkerboard masks. The $st$-networks are able to predict the masked pixels well both on in-distribution and OOD inputs from neighbouring pixels. (c), (d): RealNVP with a horizontal mask. Despite being trained on FashionMNIST, the $st$-networks are able to correctly predict the bottom half of MNIST digits in the second coupling layer due to coupling layer co-adaptation.

With the checkerboard mask, the $st$-networks predict the masked pixels from neighbouring pixels (see Appendix G for a discussion of different masks). Natural images have local structure and correlations: with a high probability, a particular pixel value will be similar to its neighbouring pixels. The checkerboard mask creates an inductive bias for the flow to pick up on these local correlations. In Figure 3, we can see that the outputs of the $s$-network are especially large for the background pixels and large patches of the same color (larger values are shown with lighter color), where the flow simply predicts for example that a pixel surrounded by black pixels would itself be black.

In addition to the checkerboard mask, RealNVP and Glow also use channel-wise masks. These masks are applied after a squeeze layer, which puts different subsampled versions of the image in different channels. As a result, the $st$-network is again trained to predict pixel values from neighbouring pixels. We provide additional visualizations for RealNVP and Glow in Appendix H.

## 6.2 Coupling layer co-adaptation

To better understand the transformations learned by the coupling layers, we replaced the standard masks in RealNVP with a sequence of *horizontal masks* that cover one half of the image (either top or bottom). For example, the first coupling layer of the flow shown in panels (c, d) of Figure 3 transforms the bottom half of the image based on the top half, the second layer transforms the top

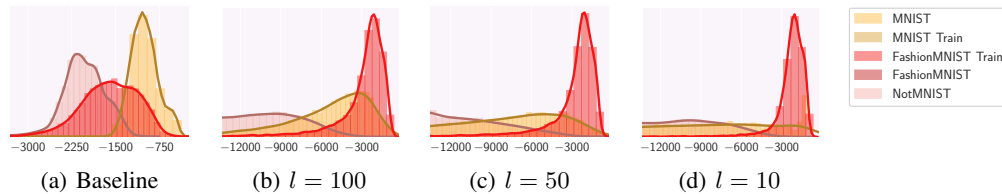

| MNIST |
| MNIST Train |
| FashionMNIST Train |
| FashionMNIST |
| NotMNIST |

(a) Baseline      (b) $l = 100$      (c) $l = 50$      (d) $l = 10$

Figure 4: **Effect of $st$-networks capacity.** Histograms of log-likelihoods of in- and out-of-distribution data for RealNVP trained on FashionMNIST, varying the dimension $l$ of the bottleneck in the $st$-networks. Flows with lower $l$ work better for OOD detection: the baseline assigns higher likelihood to the out-of-distribution MNIST images, while the flows with $l = 50$ and $l = 10$ assign much higher likelihood to in-distribution FashionMNIST data. With $l = 100$ the flow assigns higher likelihood to in-distribution data, but the overlap of the likelihood distribution with OOD MNIST is higher than for $l = 50$ and $l = 10$.

half based on the bottom half, and so on. In Figure 3(c, d) we visualize the coupling layers for a 3-layer RealNVP with horizontal masks on in-distribution (FashionMNIST) and OOD (MNIST) data.

In the first coupling layer, the shift output $-t$ of the $st$-network predicts the bottom half of the image poorly and the layer does not seem to transform the input significantly. In the second and third layer, $-t$ presents an almost ideal reconstruction of the masked part of the image on both the in-distribution and, surprisingly, the OOD input. It is not possible for the $st$-network that was only trained on FashionMNIST to predict the top half of an MNIST digit based on the other half. The resolution is that the first layer encodes information about the top half into the bottom half of the image; the second layer then decodes this information to accurately predict the top half. Similarly, the third layer leverages information about the bottom half of the image encoded by the second layer. We refer to this phenomenon as *coupling layer co-adaptation*. Additional visualizations are in Appendix H.

Horizontal masks allow us to conveniently visualize the coupling layer co-adaptation, but we hypothesize that the same mechanism applies to standard checkerboard and channel-wise masks in combination with local color correlations.

**Insights into prior work** Prior work showed that the likelihood score is heavily affected by the input complexity [39] and background statistics [35]; however, prior work does not explain *why* flows exhibit such behavior. Simpler images (e.g. SVHN compared to CIFAR-10) and background often contain large patches of the same color, which makes it easy to predict masked pixels from their neighbours and to encode and decode the information via coupling layer co-adaptation.

## 7 Changing biases in flows for better OOD detection

Our observations in Sections 5 and 6 suggest that normalizing flows are biased towards learning transformations that increase likelihood simultaneously for all structured images. We discuss two simple ways of changing the inductive biases for better OOD detection.

> By changing the masking strategy or the architecture of $st$-networks in flows we can improve OOD detection based on likelihood.

**Changing masking strategy** We consider two three types of masks. We introduced the horizontal mask in Section 6.2: in each coupling layer the flow updates the bottom half of the image based on the top half or vice versa. With a horizontal mask, flows cannot simply use the information from neighbouring pixels when predicting a given pixel, but they exhibit coupling layer co-adaptation (see Section 6.2). To combat coupling layer co-adaptation, we additionally introduce the *cycle-mask*, a masking strategy where the information about a part of the image has to travel through three coupling layers before it can be used to update the same part of the image (details in Appendix I.1). To compare the performance of the checkerboard mask, horizontal mask and cycle-mask, we construct flows of exactly the same size and architecture (RealNVP with 8 coupling layers and no squeeze layers) with each of these masks, trained on CelebA and FashionMNIST. We present the results in the Appendix I.1. As expected, for the checkerboard mask, the flow assigns higher likelihood to the

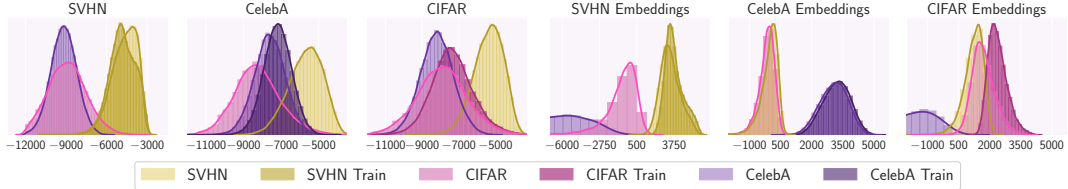

Figure 5: **Image embeddings.** Log-likelihood histograms for RealNVP trained on raw pixel data (first three panels) and embeddings extracted for the same image datasets using EfficientNet trained on ImageNet. On raw pixels, the flow assigns the highest likelihood to SVHN regardless of its training dataset. On image embeddings, flows always assign higher likelihood to in-distribution data. When trained on features capturing the semantic content of the input, flows can detect OOD.

simpler OOD datasets (SVHN for CelebA and MNIST for FashionMNIST). With the horizontal mask, the OOD data still has higher likelihood on average, but the relative ranking of the in-distribution data is improved. Finally, for the cycle-mask, on FashionMNIST the likelihood is higher compared to MNIST on average. On CelebA the likelihood is similar but slightly lower compared to SVHN.

$st$**-networks with bottleneck** Another way to force the flow to learn global structure rather than local pixel correlations and to prevent coupling layer co-adaptation is to restrict the capacity of the $st$-networks. To do so, we introduce a *bottleneck* to the $st$-networks: a pair of fully-connected layers projecting to a space of dimension $l$ and back to the original input dimension. We insert these layers after the middle layer of the $st$-network. If the latent dimension $l$ is small, the $st$-network cannot simply reproduce its input as its output, and thus cannot exploit the local pixel correlations discussed in Section 6. Passing information through multiple layers with a low-dimensional bottleneck also reduces the effect of coupling layer co-adaptation. We train a RealNVP flow varying the latent dimension $l$ on CelebA and on FashionMNIST. We present the results in Figure 4 and Appendix I. On FashionMNIST, introducing the bottleneck forces the flow to assign lower likelihood to OOD data (Figure 4). Furthermore, as we decrease $l$, the likelihood of the OOD data decreases but FashionMNIST likelihood stays the same. On CelebA the relative ranking of likelihood for in-distribution data is similarly improved when we decrease the dimension $l$ of the bottleneck, but SVHN is still assigned slightly higher likelihood than CelebA. See Appendix I for detailed results.

While the proposed modifications do not completely resolve the issue of OOD data having higher likelihood, the experiments support our observations in Section 6: preventing the flows from leveraging local color correlations and coupling layer co-adaptation, we improve the relative likelihood ranking for in-distribution data.

## 8 Out-of-distribution detection using image embeddings

In Section 4 we argued that in order to detect OOD data the model has to assign likelihood based on high-level semantic features of the data, which the flows fail to do when trained on images. In this section, we test out-of-distribution detection using image representations from a deep neural network.

> Normalizing flows can detect OOD images when trained on high-level semantic representations instead of raw pixels.

We extract embeddings for CIFAR-10, CelebA and SVHN using an EfficientNet [43] pretrained on ImageNet [37] which yields 1792-dimensional features[4]. We train RealNVP on each of the representation datasets, considering the other two datasets as OOD. We present the likelihood histograms for all datasets in Figure 5(b). Additionally, we report AUROC scores in Appendix Table 2. For the models trained on SVHN and CelebA, both OOD datasets have lower likelihood and the AUROC scores are close to 100%. For the model trained on CIFAR-10, CelebA has lower likelihood. Moreover, the likelihood distribution on SVHN, while significantly overlapping with CIFAR-10, still has a lower average: the AUROC score between CIFAR-10 and SVHN is 73%. Flows are much better at OOD detection on image embeddings than on the original image datasets. For example, a

flow trained on CelebA images assigns higher likelihood to SVHN, while a flow trained on CelebA embeddings assigns low likelihood to SVHN embeddings (see Appendix D for likelihood distribution and AUROC scores on image data).

In concurrent work, Zisselman and Tamar [49] use residual flows to approximate a distribution of intermediate layer activations of a pretrained neural network classifier and achieve strong performance in supervised out-of-distribution detection.

**Non-image data** In Appendix K we evaluate flows on tabular UCI datasets, where the features are relatively high-level compared to images. On these datasets, normalizing flows assign higher likelihood to in-distribution data.

## 9 Conclusion

Many of the puzzling phenomena in deep learning can be boiled down to a matter of *inductive biases*. Neural networks in many cases have the flexibility to overfit datasets, but they do not because the biases of the architecture and training procedures can guide us towards reasonable solutions. In performing OOD detection, the biases of normalizing flows can be more of a curse than a blessing. Indeed, we have shown that flows tend to learn representations that achieve high likelihood through generic graphical features and local pixel correlations, rather than discovering semantic structure that would be specific to the training distribution.

While we show that flows tend to focus on low-level features of the image rather than its semantic content when assigning likelihoods, flows are often able to produce samples semantically similar to the training data. To the best of our knowledge, the question of how the flows are able to produce these samples and to what extent they suffer from memorization of the training data has not been thoroughly studied in the literature. We leave a careful study of the sampling in normalizing flows as an exciting direction for future work.

To provide insights into prior results [e.g., 29, 7, 30, 40, 48, 39], part of our discussion has focused on an in-depth exploration of the popular class of normalizing flows based on affine coupling layers. We hypothesize that many of our conclusions about coupling layers extend at a high level to other types of normalizing flows [e.g., 3, 6, 12, 21, 14, 32, 41, 18, 8]. A full study of these other types of flows is a promising direction for future work.

## 10 Broader impact

Out-of-distribution detection is crucial for robust, reliable and fair machine learning systems. Mitchell et al. [27] and Gebru et al. [13] argue that applying machine learning models outside of the context where they were trained and tested can lead to dangerous and discriminatory outcomes in high-stake domains. We hope that our work will generally contribute to the understanding of out-of-distribution detection and facilitate methodological progress in this area.

**Acknowledgements**

This research is supported by an Amazon Research Award, Facebook Research, Amazon Machine Learning Research Award, NSF I-DISRE 193471, NIH R01 DA048764-01A1, NSF IIS-1910266, and NSF 1922658 NRT-HDR: FUTURE Foundations, Translation, and Responsibility for Data Science. We thank Marc Finzi, Greg Benton, Wesley Maddox, and Alex Wang for helpful discussions.

## Footnotes

[2]For the details of the visualization procedure and the training setup please see Appendices E and C.

[3]When training flow models on images or other discrete data, we use dequantization to avoid pathological solutions [46, 44]: we add uniform noise $\epsilon \sim U[0; 1]$ to each pixel $x_i \in \{0, 1, \ldots, 255\}$. Every time we pass an image through the flow $f(\cdot)$, the resulting latent representation $z$ will be different.

[4]The original images are 3072-dimensional, so the dimension of the embeddings is only two times smaller. Thus, the inability to detect OOD images *cannot* be explained just by the high dimensionality of the data.

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
