[Supplementary Material]



Figure 6: **Inductive biases define what data is OOD.** A conceptual visualization of two distributions in the image space (shown in yellow and red), training CelebA data is shown with crosses, and other images are shown with circles. The distribution shown in yellow could represent inductive biases of a human: it assigns high likelihood to all images of human faces, regardless of small levels of noise, and small brightness changes. The second distribution, shown in red, could represent a normalizing flow: it assigns high likelihood to all smooth structured images, including images from SVHN and ImageNet. Both distributions assign the same likelihood to the training set, but their high-probability sets are different.

## Appendix outline

This appendix is organized as follows.

## A   Maximum likelihood objective is agnostic to what data is OOD

In Section 4 we argued that the maximum likelihood objective by itself does not define out-of-distribution detection preformance of a normalizing flow. Instead, it is the inductive biases of the flow that define what data will be assigned with high or low likelihood. We illustrate this point in Figure 6.

The yellow and red shaded regions illustrate the high-probability regions of two distributions defined on the image space $\mathcal{X}$. The distribution in yellow assigns high likelihood to the train (CelebA) images

corrupted by a small level of noise, or brightness adjustments. This distribution represents how a human could describe the target dataset. The red distribution on the other hand assigns high likelihood to all structured images including those from ImageNet and SVHN, but does not support noisy train images. The red distribution represents a distribution learned by normalizing flow.

For simplicity, we could think that the distributions are uniform on the highlighted sets, and the sets have the same volume. Then, both distributions assign equally high likelihood to the training data, but the split of the data into in-distribution and OOD is different. As both distributions provide the same density to the target data, the value of the maximum likelihood objective in Equation (1) would be the same for the corresponding models.

More generally, for any distribution that only assigns finite density to the train set, we can construct another distribution that assigns the same density to the train data, but also high density to a given set of (OOD) datapoints. In particular, the new distribution will achieve the same value of the maximum likeihood objective in Equation (1). We formalize our reasoning in the following simple proposition.

**Proposition 1.** *Let $p(\cdot)$ be a probability density on the space $\mathcal{X}$, and let $\mathcal{D} = \{x_i\}_{i=1}^{N}$ be the training dataset, where $x_i \in \mathcal{X}$ for $i = 1, \ldots, N$. Assume for simplicity that $p$ is upper bounded: for any $x$ $p(x) \leq u$. Let $\mathcal{D}_{OOD}$ be an arbitraty finite set of points. Then, for any $c \geq 0$ there exists a distribution with density $p'(\cdot)$ such that $p'(x) = p(x)$ for all $x \in \mathcal{D}$, and $p'(x') \geq c$ for all $x' \in \mathcal{D}_{OOD}$.*

*Proof.* Consider the set $\mathcal{S}(r) = \cup_{x_i \in \mathcal{D}} B(x_i, r)$, where $B(x, r)$ is a ball of radius $r$ centered at $x$. The probability mass of this set $P(\mathcal{S}(r)) = \int_{x \in \mathcal{S}(r)} p(x)dx$. As $r \to 0$, the volume $V(\mathcal{S}(r))$ of the set $\mathcal{S}(r)$ goes to zero. We have

$$P(\mathcal{S}(r)) = \int_{x \in \mathcal{S}(r)} p(x)dx \leq V(\mathcal{S}(r)) \cdot u \xrightarrow{r \to 0} 0. \tag{4}$$

Hence, there exists $r_0$ such that $P(\mathcal{S}(r_0)) \leq \frac{1}{2}$.

Now, define the a neighborhood of the set $\mathcal{D}_{\text{OOD}}$ as

$$\mathcal{S}_{\text{OOD}} = \cup_{x' \in \mathcal{D}_{\text{OOD}}} B(x, \hat{r}), \tag{5}$$

where $\hat{r}$ is selected so that the total volume of set $\mathcal{S}_{\text{OOD}}$ is $1/2c$. Then, we can define a new density $p'$ by redistributing the mass in $p(\cdot)$ from outside the set $\mathcal{S}(r_0)$ to the neighborhood $\mathcal{S}_{\text{OOD}}$ as follows:

$$p'(x) = \begin{cases} p(x), & \text{if } x \in \mathcal{S}(r_0), \\ 2c \cdot \big(1 - P(\mathcal{S}(r_0))\big), & \text{if } x \in \mathcal{S}_{\text{OOD}}, \\ 0, & \text{otherwise.} \end{cases} \tag{6}$$

The density $p'(\cdot)$ integrates to one, coincides with $p$ on the training data, and assigns density of at least $c$ to points in $\mathcal{D}_{\text{OOD}}$. $\qquad\square$

# B    Flows have capacity to distinguish datasets

Normalizing flows are unable to detect OOD image data when trained to maximize likelihood on the train set. It is natural to ask whether these models are at all capable of distinguishing different image datasets. In this section we demonstrate the following:

> **Observation**: Flows can assign high likelihood to the train data and low likelihood to a given OOD dataset if they are explicitly trained to do so.
> **Relevance to OOD detection**: While flows have sufficient capacity to distinguish different data, they are biased towards learning solutions that assign high likelihood to all structured data and consequently fail to detect OOD inputs.

We introduce an objective that encouraged the flow to maximize likelihood on the target dataset and to minimize likelihood on a specific OOD dataset. The objective we used is

$$\frac{1}{N_{\mathcal{D}}} \sum_{x \in \mathcal{D}} \log p(x) - \frac{1}{N_{\text{OOD}}} \sum_{x \in \mathcal{D}_{\text{OOD}}} \log p(x) \cdot I[\log p(x) > c], \tag{7}$$

Figure 7: **Negative training.** The histograms of log-likelihood for RealNVP when in training likelihood is maximized on one dataset and minimized on another dataset: (a) maximized on CIFAR, minimized on SVHN; (b) maximized on SVHN, minimized on CIFAR; (c) maximized on CIFAR, minimized on CelebA; (d) maximized on CelebA, minimized on CIFAR. (e) maximized on FashionMNIST, minimized on MNIST; (f) maximized on MNIST, minimized on FashionMNIST;

where $I[\cdot]$ is an indicator function and the constant $c$ allows us to encourage the flow to only push the likelihood of OOD data to a threshold rather than decreasing it to $-\infty$; $N_{\mathcal{D}}$ is the number of train datapoints and $N_{\text{OOD}} = \sum_{x \in \mathcal{D}_{\text{OOD}}} I[\log p(x) > c]$ is the number of OOD datapoints that have likelihood above the threshold $c$.

We trained a RealNVP flow with the objective (7) using different pairs of target and OOD datasets: CIFAR-10 vs CelebA, CIFAR-10 vs SVHN and FashionMNIST vs MNIST. We present the results in Figure 7. In each case, the flow is able to push the likelihood of the OOD dataset to very low values, and simultaneously maximize the likelihood on the target dataset creating a clear separation between the two.

**Hyper-parameters** For the flow architecture and training used the same hyper-parameters as we did for the baselines, described in Appendix C. For CelebA, CIFAR and SVHN models we set $c = -100000$, and for MNIST, FashionMNIST and NotMNIST we set $c = -30000$.

**Connection with prior work** Flows can be used as classifiers separating different classes of the same dataset [30, 20, 1], which further highlights the fact that flows can distinguish images based on their contents when trained to do so. A similar experiment for the PixelCNN model [31] was presented in Hendrycks et al. [15]. The authors maximized the likelihood of CIFAR-10 and minimized the likelihood of the TinyImages dataset [45]. In their experiments, this procedure consistently led to CIFAR-10 having higher likelihood than any of the other benchmark datasets. In Figures 7, for each experiment in addition to the two datasets that were used in training we show the log-likelihood distribution on another OOD dataset. For example, when we train the flow to separate CIFAR-10 from CelebA (panels c, d), the flow successfully does so but assigns SVHN with likelihood similar to that of CIFAR. When we train the flow to separate CIFAR-10 from SVHN (panels c, d), the flow successfully does so but assigns CelebA with likelihood similar to that of CIFAR. Similar observations can be made for MNIST, FashionMNIST and notMNIST. At least for normalizing flows, minimizing the likelihood on a single OOD dataset does not lead to all the other OOD datasets achieving low-likelihood.

## C   Details of the experiments

**RealNVP** For all RealNVP models, we generally follow the architecture design of Dinh et al. [10]. We use multi-scale architecture where after a block of coupling layers half of the variables

Figure 8: **Baseline log-likelihoods.** The histograms of log-likelihood for RealNVP and Glow models trained on various datasets. Both flows consistently assign similar or higher likelihood to OOD data compared to the target dataset. The likelihood distribution for train and test sets of the target data is typically very similar.

are factored out and copied forward directly to the latent representation. Each scale consists of 3 coupling layers with checkerboard mask, followed by a squeeze operation and 3 coupling layers with channel-wise mask (see Figure 9). For the $st$-network we use deep convolutional residual networks with additional skip connections following Dinh et al. [10]. In all experiments, we use Adam optimizer. On grayscale images (MNIST, FashionMNIST), we used 2 scales in RealNVP, 6 blocks in residual $st$-network, learning rate $5 \times 10^{-5}$, batch size 32 and trained model for 80 epochs. On CIFAR-10, CelebA and SVHN, we used 3 scales, 8 blocks in $st$-network, learning rate $10^{-4}$, batch size 32, weight decay $5 \times 10^{-5}$ and trained the model for 100 epochs. On ImageNet, we used 5 scales, 2 blocks in $st$-network, learning rate $10^{-3}$, batch size 64, weight decay $5 \times 10^{-5}$ and trained the model for 42 epochs. On CelebA $64 \times 64$, we used 4 scales, 4 blocks in $st$-network, learning rate $10^{-4}$, batch size 64, weight decay $5 \times 10^{-5}$ and trained the model for 100 epochs.

**Glow**   We follow the training details of Nalisnick et al. [29] for multi-scale Glow models. Each scale consists of a sequence of actnorm, invertible $1 \times 1$ convolution and coupling layers [23]. The squeeze operation is applied before each scale, and half of the variables are factored out after each scale. In all experiments, we use RMSprop optimizer. On grayscale images (MNIST, FashionMNIST), we used 2 scales with 16 coupling layers, a 3-layer Highway network with 200 hidden units for $st$-network, learning rate $5 \times 10^{-5}$, batch size 32 and trained model for 80 epochs. On color images

(a) Squeeze layer      (b) CB mask    (c) CW mask    (d) Hor. mask

Figure 9: **Squeeze layers and masks. (a)**: A squeeze layer squeezes an image of size $c \times h \times w$ into $4c \times h/2 \times w/2$. The first panel shows the mask, where each color corresponds to a channel added by the squeeze layer (for visual clarity we show the mask for a $12 \times 12$ image). The second panel shows a $1 \times 28 \times 28$ MNIST digit, and the last panel shows the 4 channels produced by the squeeze layer. The colors of the boundaries of the channel visualizations correspond to the colors of the pixels in the mask. Each channel produced by the squeeze layer is a subsampled version of the input image. **(b)-(d)**: Checkerboard, channel-wise and horizontal masks applied to the same input image. Masked regions are shown in red. Channel-wise mask is obtained by applying a squeeze layer and masking two of the channels (e.g. the last two); here we show the masked pixels in the un-squeezed image. Masks are typically alternated: in the subsequent layers the masked and observed positions are swapped.

(CIFAR-10, CelebA, SVHN), we used 3 scales with 8 coupling layers, a 3-layer Highway network with 400 hidden units for $st$-network, learning rate $5 \times 10^{-5}$, batch size 32 and trained model for 80 epochs.

## D    Baseline models likelihood distributions and AUROC scores

In Figure 8, we plot the histograms of the log likelihoods on in-distribution and out-of-distribution datasets RealNVP and Glow models. In Table 1 we report AUROC scores for OOD detection with these models. As reported in prior work, Glow and RealNVP consistently fail at OOD detection.

| Model | Train data | OOD data | | | | OOD data | | |
|---|---|---|---|---|---|---|---|---|
| | | CelebA | CIFAR-10 | Data | SVHN | MNIST | Fashion | NotMNIST |
| RealNVP | CelebA | – | 67.7 | 6.3 | MNIST | – | 99.99 | 99.99 |
| | CIFAR-10 | 56.0 | – | 6.0 | Fashion | 10.8 | – | 72.1 |
| | SVHN | 99.0 | 98.4 | – | | | | |
| Glow | CelebA | – | 69.1 | 6.4 | MNIST | – | 99.96 | 100.0 |
| | CIFAR-10 | 52.9 | – | 5.5 | Fashion | 13.3 | – | 80.2 |
| | SVHN | 99.9 | 99.1 | – | | | | |

Table 1: **Baseline AUROC.** AUROC scores on OOD detection for RealNVP and Glow models trained on various image data. Flows consistently assign higher likelihoods to OOD dataset except when trained on MNIST and SVHN. The AUROC scores for RealNVP and Glow are close.

## E    Visualization implementation

Normalizing flows such as RealNVP and Glow consist of a sequence of coupling layers which change the content of the input and squeeze layers (see Figure 9) which reshape it. Due to the presence of squeeze layers, the latent representations of the flow have a different shape compared to the input. In order to visualize latent representations, we revert all squeezing operations of the flow and visualize `unsqueeze(z)`. Similarly, for visualization of coupling layer activations and scale and shift parameters predicted by $st$-network, we revert all squeezing operations and join all factored out tensors in the case of multi-scale architecture (i.e., we feed the corresponding tensor through inverse sub-flow without applying coupling layers or invertible convolutions).

(a) RealNVP trained on FashionMNIST

(b) Glow trained on FashionMNIST

(c) RealNVP trained on CelebA

Figure 10: **Latent spaces.** Visualization of latent representations for RealNVP and Glow models on in-distribution and out-of-distribution inputs. **Rows 1-3 in (a) and (b)**: original images, latent representations, latent representation averaged over 40 samples of dequantization noise for RealNVP and Glow model trained on FashionMNIST and using MNIST for OOD data. **Row 4 in (a)**: latent representations for batch normalization in train mode. **Rows 1-4 in (c)**: original images, latent representations, the blue channel of the latent representation, and the latent representations for batch normalization in train mode for a RealNVP model trained on CelebA and using SVHN as OOD data. For both dataset pairs, we can recognize the shape of the input image in the latent representations. The flow represents images based on their graphical appearance rather than semantic content.

## F  Additional latent representation visualizations

in Figure 10, we plot additional latent representations for RealNVP and Glow trained on FashionM-NIST with MNIST as OOD dataset, RealNVP trained on CelebA with SVHN as OOD. The results agree with Section 5: we can recognize edges from the original inputs in their latent representations.

(a) RealNVP trained on FashionMNIST

(b) Glow trained on FashionMNIST

(c) RealNVP trained on CelebA

Figure 11: **Coupling layer visualizations.** Visualization of intermediate coupling layer activations and $st$-network predictions for **(a)**: RealNVP trained on FashionMNIST; **(b)**: Glow trained on FashionMNIST; **(c)**: RealNVP trained on CelebA. The top half of each subfigure shows the visualizations for an in-distribution image (FashionMNIST or CelebA) while the bottom half shows the visualizations for an OOD image (MNIST or SVHN). For all models, the shape of the input both for in- and out-of-distribution image is clearly visible in $s$ and $t$ predictions of the coupling layers.

### F.1 Receptive fields of latent representations

To further analyze latent representations learned by normalizing flow, we study the receptive fields of each coordinate in the latent representation. To do so, we compute and visualize the Jacobian $\frac{dz_i}{dx}$ for different latent coordinates $z_i$ of an in-distribution FashionMNIST image and show the visulization for two coordinates in Figure 12. For most latent coordinates the receptive field is limited to the neighbouring pixels as we show in Fig. 12 left. However, coordinates $z_i$ corresponding to the edges in the input image are weakly affected by longer range dependencies from other pixels near the edges (Fig. 12 right).

Figure 12: **Receptive fields of latent representations**. Visualization of the Jacobian $\frac{dz_i}{dx}$ for two different coordinates in the latent space. We use RealNVP trained on FashionMNIST and the latent representations are computed for an in-distribution input.

## G Masking strategies

In Figure 9, we visualize checkerboard, channel-wise masks and horizontal masks on a single-channel image. The checkerboard and channel-wise masks are commonly used in RealNVP, Glow and other coupling layer-based flows for image data. We use the horizontal mask to better understand the transformations learned by the coupling layers in Section 6.

## H Additional coupling layer visualizations

In Figure 11, we plot additional visualizations of coupling layer activations and scale $s$ and shift $t$ parameters predicted by $st$-networks. In Figure 13 we visualize the coupling layer activations for the small flow with horizontal mask from Section 6.2 on several additional OOD inputs. These visualizations provide additional empirical support for Section 6.

Figure 13: **Coupling layer co-adaptation.** Visualization of intermediate coupling layer activations, as well as scales $s$ and shifts $t$ predicted by each coupling layer of a RealNVP model with a horizontal mask on out-of-distribution MNIST inputs. Although RealNVP was trained on FashionMNIST, the $st$-networks are able to correctly predict the bottom half of MNIST digits in the second coupling layer due to coupling layer co-adaptation.

## I Changing biases in flow models for better OOD detection

### I.1 Cycle-mask

In Section 6 we identified two mechanisms through which normalizing flows learn to predict masked pixels from observed pixels on OOD data: leveraging local color correlations and coupling layer co-adaptation. We reduce the applicability of these mechanisms with *cycle-mask*: a new masking strategy for the coupling layers illustrated in Figure 14.

Figure 14: **Cycle-mask.** A new sequence of masks for coupling layers in RealNVP that we evaluate in Section 7. We separate the input image space of size $c \times h \times w$ into four quadrants of size $c \times h/2 \times w/2$ each. Each coupling layer transforms one quadrant based on the previous quadrant. Cycle-mask prevents co-adaptation between subsequent coupling layers discussed in Section 6: the information from a quadrant has to propagate through four coupling layers before reaching the same quadrant.

(a) Checkerboard Mask　　(b) Horizontal Mask　　(c) Cycle-Mask

(d) Checkerboard Mask　　(e) Horizontal Mask　　(f) Cycle-Mask

Figure 15: **Effect of masking strategy** The first two rows show log likelihood distribution for RealNVP models trained on FashionMNIST and CelebA with (a) checkerboard mask; (b) horizontal mask; and (c) cycle-mask. The third and the fourth rows show samples produced by the corresponding models.

Figure 16: **Effect of $st$-network capacity. The first row** shows the histogram of log likelihoods for a RealNVP model trained on CelebA dataset: **(a)** for a baseline model, and **(b)-(d)** for models with different bottleneck dimensions $l$ in $st$-network. **The second and third rows** show samples from RealNVP model trained on CelebA and FashionMNIST respectively: **(e)** and **(i)** for baseline models, and **(f)-(h)** and **(j)-(l)** for models with different bottleneck dimensions $l$. In **(m)**, we show the visualization of the coupling layer activations and $st$-network predictions for a RealNVP model trained on FashionMNIST with a bottleneck of dimension $l = 10$. The top half shows the visualizations for an in-distribution FashionMNIST image while the bottom half shows the visualizations for an OOD MNIST image. $st$-network with restricted capacity cannot accurately predict masked pixels of the OOD image in the intermediate coupling layers. Moreover, in the middle coupling layers for the MNIST input the activations resemble FashionMNIST images in $s$ and $t$ predictions.

With cycle-mask, the coupling layers do not have access to neighbouring pixels when predicting the masked pixels, similarly to the horizontal mask. Furthermore, cycle mask reduce the effect of coupling layer co-adaptation: the information about a part of the image has to travel through $4$ coupling layers before it can be used to update the same part of the image.

**Changing masking strategy** In Figure 15 we show the log-likelihood histograms and samples for a RealNVP of a fixed size with checkerboard, horizontal and cycle-mask.

**Changing the architecture of $st$-networks** In Figure 16, we show likelihood distributions, samples and coupling layer visualization for RealNVP model with $st$-network with a bottleneck trained on FashionMNIST and CelebA datasets. The considered bottleneck dimensions for FashionMNIST are $\{10, 50, 100\}$, and for CelebA the dimensions are $\{30, 80, 150\}$. In the baseline RealNVP model, we use a standard deep convolutional residual network without additional skip connections from the intermediate layers to the output which were used in Dinh et al. [10].

# J  Samples

In Figure 18, we show samples for RealNVP and Glow models trained on CelebA, CIFAR-10, SVHN, FashionMNIST and MNIST, and a RealNVP model trained on ImageNet $64 \times 64$ and CelebA $64 \times 64$.

## J.1  Latent variable resampling

To further understand the structure of the latent representations learned by the flow, we study the effect of resampling part of the latent representations corresponding to images from different datasets from the base Gaussian distribution. In Figure 17, using a RealNVP model trained on CelebA we compute the latent representations corresponding to input images from CelebA, SVHN, and CIFAR-10 datasets, and randomly re-sample the subset of latent variables corresponding to a $10 \times 10$ square in the center of the image (to find the corresponding latent variables we apply the squeeze layers from the flow to the $32 \times 32$ mask). We then invert the flow and compute the reconstructed images from the altered latent representations.

Both for in-distribution and out-of-distribution data, the model almost ideally preserves the part of the image other than the center, confirming the alignment between the latent space and the original input space discussed in Section 5. The model adds a face to the resampled part of the image, preserving the consistency with the background to some extent.

(a) Celeb-A         (b) CIFAR-10         (c) SVHN

Figure 17: **Latent variable resampling.** Original images (**top row**) and reconstructions with the latent variables corresponding to a $10 \times 10$ square in the center of the image randomly re-sampled for a RealNVP model trained on Celeb-A (**bottom row**). The model adds faces (as it was trained Celeb-A) to the part of the image that is being re-sampled.

| (a) RNVP, CelebA | (b) RNVP, CIFAR-10 | (c) RNVP, SVHN | (d) RNVP, FashionM-NIST |
| (e) RNVP, MNIST | (f) RNVP, CelebA-HQ | (g) RNVP, ImageNet | (h) Glow, CelebA |
| (i) Glow, CIFAR-10 | (j) Glow, SVHN | (k) Glow, Fashion | (l) Glow, MNIST |

Figure 18: **Baseline Samples.** Samples from baseline RealNVP and Glow models. For ImageNet and CelebA-HQ we used datasets with $(64 \times 64)$ definition.

## K  Out-of-distribution detection on tabular data

<div align="center">(a) Image embeddings</div>

| Train data | OOD data | | |
|---|---|---|---|
| | CelebA | CIFAR-10 | SVHN |
| CelebA | – | 99.99 | 99.99 |
| CIFAR-10 | 99.99 | – | 73.31 |
| SVHN | 100.0 | 99.98 | – |

<div align="center">(b) Tabular data</div>

| Train class (OOD class) | Dataset | |
|---|---|---|
| | HEPMASS | MINIBOONE |
| Background (Signal) | 83.78 | 72.71 |
| Signal (Background) | 70.73 | 87.56 |

Table 2: **Image embedding and UCI AUROC. (a)**: AUROC scores on OOD detection for RealNVP model trained on image embeddings extracted from EfficientNet. The model is trained on one of the embedding datasets while the remaining two are considered OOD. The models consistently assign higher likelihood to in-distribution data, and in particular AUROC scores are significantly better compared to flows trained on the original images (see Table 1). **(b)**: AUROC scores on OOD detection for RealNVP trained on one class of Hepmass and Miniboone datasets while the other class is treated as OOD data.

### K.1  Model

We use RealNVP with 8 coupling layers, fully-connected $st$-network and masks which split input vector by half in an alternating manner. For UCI experiments, we use 1 hidden layer and 256 hidden units in $st$-networks, learning rate $10^{-4}$, batch size 32 and train the model for 100 epochs. For image

<div style="text-align:center">(a) Miniboone Dataset        (b) Hepmass Dataset</div>

Figure 19: **UCI datasets.** The histograms of log-likelihood for RealNVP on Hepmass and Miniboone tabular datasets when trained on one class and the other class is viewed as OOD. The train and test likelihood distributions are almost identical when trained on either class, and the OOD class receives lower likelihoods on average. There is however a significant overlap between the likelihoods for in- and out-of-distribution data.

embeddings experiments, we use 3 hidden layer and 512 hidden units in $st$-networks, learning rate $10^{-3}$, batch size 1024 and train the model for 120 epochs. For all experiments, we use the AdamW optimizer [25] and weight decay $10^{-3}$.

## K.2    EfficientNet embeddings

We train RealNVP model on image embeddings for CIFAR-10, CelebA and SVHN extracted from EfficientNet train on ImageNet, and report AUROC scores in Table 2(a).

## K.3    UCI datasets

In this experiment, we use 2 UCI classification datasets which were used for unsupervised modeling in prior works on normalizing flows [32, 11, 14]: HEPMASS [2] and MINIBOONE [36]. HEPMASS and MINIBOONE are both binary classification datasets originating from physics, and the two classes represent *background* and *signal*. We follow data preprocessing steps of Papamakarios et al. [32]. We filter features which have too many reoccurring values, after that the dimenionality of the data is 15 for HEPMASS and 50 for MINIBOONE. For HEPMASS, we use the "1000" dataset which contains subset of particle signal with mass 1000. For MINIBOONE data, for each class we take a random split of 10% for a test set.

To test OOD detection performance, for each dataset we train a model on one class while treating the second class as OOD data. We plot the resulting train, test and OOD likelihood distributions for each dataset in Figure 19. We also report AUROC scores for each setup in Table 2(b). While test and OOD likelihoods overlap, the in-distribution class has higher average likelihood in all cases, and AUROC values are ranging between 70% and 87% which is a significantly better result compared to the results for image benchmarks reported in Nalisnick et al. [29].