[Reviews · NeurIPS 2020]

Review 1

Summary and Contributions: ----- Update ----- I have read the author response as well as the other reviews. I agree with some of the concerns raised by the other reviewers, in particular that some of the conclusions drawn in the paper are too general. The authors seem to agree on this in their response as well and declared to correct this in a final version. Overall, I still find this work provides valuable insights into the inductive biases of NFs and their ramifications for OOD. I lowered my score to 7, but still would vote for accepting this work. ------------------ This work analyzes the phenomenon that deep generative models (DGMs) regularly assign higher likelihood to out-of-distribution (OOD) samples, focusing in particular on the model class of normalizing flows (NFs). The analysis concludes that the inductive biases of NFs, specifically the coupling layer mechanism used in many models to achieve invertibility (e.g. NICE, RealNVP, Glow, etc.), is a major culprit which causes NFs to learn latent representations that are based on low-level texture correlations rather than high-level semantic content. Two reasons for this issue of the coupling layers are presented: (1) the commonly used checkerboard masking causing models to exploit local pixel correlations (nearby pixels likely share similar values), and (2) the information of masked pixels is locally encoded in subsequent layers ('coupling layer co-adaptation'). Since these low-level statistics are generic to natural, structured images, all such images (whether in- or out-of-distribution w.r.t. semantics) are assigned high likelihood under the model. Two solutions to adapt these biases are presented: (1) cycle-masking which prevents co-adaptation, and (2) introducing bottlenecks into the scale and translation networks, which both show marked improvements in detection performance on common OOD benchmarks (MNIST, Fashion-MNIST, notMNIST, SVHN, CelebA, CIFAR-10 in-/out-of-distribution combinations). Finally, an experiment with embeddings from a network pre-trained on ImageNet is presented to indicate that NFs can detect OOD images when trained on high-level, semantic features.

Strengths: - The paper presents a technically sound analysis on an important issue that is relevant from both a scientific (understanding DGMs) as well as a practical perspective (robustness and trustworthiness in applications). - The arguments are presented clearly and the experiments to verify the claims are well designed and scientifically sound (e.g. using horizontal masks to test for coupling layer co-adaptation). - The analysis highlights the importance of network architecture inductive biases for OOD generalization. - Two possible solutions (cycle mask and bottlenecks) to adapt the coupling layer inductive biases are presented which yield an improvement in OOD detection performance.

Weaknesses: - The implications of the presented insights and analysis, focusing on coupling layer inductive biases of NFs, are unclear for the general class of DGMs (e.g. VAEs or autoregressive models) for which the OOD phenomenon has also been observed [1]. I see that this will also be subject to future work, but are there specific connections/implications that apply to other models as well? Highlighting some of these would greatly improve the value of this work (e.g. a tendency towards local pixel statistics, etc.). A VAE usually has a latent hierarchy and bottleneck (one of the solutions), but suffers from the same issue [1]. - I find the added value of Section 8 rather minor, i.e. it is not surprising that NFs preserve semantic features when these have been learned *a priori* by some supervised model. The challenging question, of course, is how to learn such high-level semantic features without (or very few) labelled information. Did I maybe miss something? ##### [1] E. Nalisnick, A. Matsukawa, Y. W. Teh, D. Gorur, and B. Lakshminarayanan. Do deep generative models know what they don’t know? In ICLR, 2018.

Correctness: - The analysis is technically sound and the experimental evaluation scientifically rigorous. - The experimental design is well thought out to test the raised hypotheses (e.g. applying horizontal masking).

Clarity: - The paper is well structured and written clearly. - Highlighting the key messages in these blue boxes adds to the structure, but gets somewhat repetitive in a paper with only 8 pages in my opinion.

Relation to Prior Work: - The paper places the work well into the related literature up to some very recent works. - The presented findings are explicitly related to explain observations of previous works (e.g. explaining the presence of latent correlations that group anomaly detection methods exploit). - Section 2 provides a good, concise summary and background on normalizing flows.

Reproducibility: Yes

Additional Feedback: - At least a brief explanation in the main paper about the visualization procedure used would be good to get an intuition of what is shown. - I found your intuition on coupling layers (first paragraph in Section 6) helpful.


Review 2

Summary and Contributions: This paper performs an in-depth empirical analysis of RNVP-style normalizing flows applied to image data. The first experiment examines images (both in- and out-of-distribution) as they are passed through the flow. The paper finds that even the OOD data is only lightly distorted as it progresses through the transforms. Secondly, it is observed that coupling layers aim to predict their input, and this is most easily done by using local pixel correlations. Thirdly, two strategies are shown to mitigate high OOD likelihoods. The first is to use a cycle masking strategy, based on the insights about coupling layer co-adaptation. The second is to train on a higher-level representation, such as an embedding or tabular data. In this case, the in-distribution set has higher likelihood. This further supports the hypothesis that low-level pixel statistics are contributing to the OOD phenomenon.

Strengths: I very much enjoyed this paper: it is clear, interesting, and novel. The juggling of high-level narrative while incorporating enough supporting details is commendable. Some things I liked in particular… Deep-dive into the coupling layer: Coupling layers---as used in RNVP and Glow---are the backbone of flow-based generative modeling. This paper performs an in-depth analysis, pointing out some fundamental facts that, in hindsight, should really have been discussed earlier. I think this paper is essential reading for anyone using RNVP / Glow-style architectures. Insights into prior work: I really like how this paper makes connections to and explanations for previous work. For instance, the connections to previous anomaly detection methods (line 151). For another example, the observation about local pixel correlations perfectly explains the high likelihood of constant images (line 208). Nice visualizations: I found the paper’s visualizations beautifully done as well as informative. While the paper’s results are somewhat unorthodox, leaning more towards the qualitative, I found them to convincingly support their arguments.

Weaknesses: I have only two criticisms, one moderate and one minor. Assumption that likelihood is good for OOD detection (lines 60, 99): The paper seems to assume that likelihood / probability density alone should be able to detect OOD inputs. While I agree that, in practice, we can expect it to in many situations, there are clear counter examples. The most notable follows from the Gaussian annulus theorem, as pointed out by [Choi et al. 2018, Nalisnick et al. 2019]: points near the mode are not in the Gaussian model’s typical (or high probability) set. This should at least be acknowledged in the paper. Results somewhat limited to coupling layers: This paper is somewhat limited in scope in its focus on coupling layers. Yet, I consider this a minor issue given (1) the popularity of RNVP and Glow and (2) the focus is necessary to perform the in-depth investigation that they did.

Correctness: Yes, I did not find any errors of note.

Clarity: Yes, as mentioned above, the paper does an excellent job of conveying a narrative arc through a series of nice visualizations. While many of the details have been pushed to the appendix, I think this is unavoidable given the page constraints.

Relation to Prior Work: Yes

Reproducibility: Yes

Additional Feedback: POST REBUTTAL UPDATE: Thank you, authors, for responding to our reviews. After considering the other reviews and the author response, I will leave my score unchanged at 7.


Review 3

Summary and Contributions: This paper concerns the ongoing difficulties in detecting out-of-distribution images using likelihood models. It provides various experiments giving insights into the issues as well as offering several possibilities how to avoid them.

Strengths: Flow-based likelihood models in general have seen increasing research interest recently. Better understanding their internal workings and inductive biases is a key ingredient of advancing the field, and therefore of high significance. Furthermore, OoD detection is a critical component for trustworthy and reliable AI. The co-adaptation strategy of the first few coupling blocks in a flow-network is an extremely interesting observation in its own right. The regularization techniques using alternative masking schemes, or a bottleneck in the st-network seem of high value to the community; either used directly, or as starting points for even more advanced ways of controlling the inductive bias of flow-based models.

Weaknesses: While I enjoyed reading about the various experiments, I do not think the hypotheses were confirmed in a rigorous way, and some aspects might not be correct. I fear this may lead to some false assumptions and misconceptions in the field going forward, so I think it could be overall damaging or a step backwards for this paper to appear at NeurIPS in its current form. I am not saying the hypotheses and speculations are necessarily false, but I do feel like they are falsely portrayed as proven facts at various points, while not being adequately confirmed by experiment. ------ 1.) At several points, it is implied or stated that the flows do not learn the semantic content of the inputs, and only learn certain local structures (l.29-31; l.133,end; l.277,278). But I think several observations stand in conflict with this: a) Appendix Fig. 14, 15, 16: The model clearly *does* generate images with the same semantic content as the inliers (e.g. Faces, digits, clothes). If the statement were true, the model would generate images that only have the same low-level correlations but not the semantic content. b) Reference [38]: When subtracting the log-likelihood of a second model that is forced to ignore semantics through data augmentation and *only* model local details, the OoD-detection works very well. Again this indicates that the standard flow learns a correct model of the semantics, but the pure log-likelihood does not quantify them in a useful way. As others have pointed out in the past, the semantic content contains perhaps 20 bits of information, and the pixel values contain 20000 bits. So if the pure log-likelihood is used for OoD detection, even a one-per-mille variance in the image statistics or local correlations drowns out the semantic OoD signal. But this does not say anything about whether the semantics are learned correctly or not, simply that the likelihood is not a useful score to quantify OoD (see [7],[29],[30] for further discussion). Also see e.g. Sec. 3.1, 3.2 in Fetaya et al, "Understanding the Limitations of Conditional Generative Models", although they make the semantic information explicit. ------ 2.) I take particular issue with the blue box of l.133 and Sec. 5 overall. - Why does a correspondence between latent- and image-coordinates imply that the latent space *only* encodes visual appearance? Perhaps the semantics are globally encoded and therefore not clearly visible (e.g. the weak structures throughout the latent images in the left half of Fig. 2). Again, Figs. 14-16 imply that the semantics are encoded somewhere in latent space. - How can we conclude anything from this about the OoD score? [29] showed that the Jacobian term accounts for a large part of the log-likelihood alongside the p(z) term, and primarily decides over the OoD decision for the SVHN/CIFAR case. - I do not believe averaging the dequantization actually provides any useful information, and I don't think those images can be used to make any factual statements. It is not clear what is going on in the process: For instance, if the semantic image content were encoded globally, but modulated with the local noise in the empty regions, the averaged latent output would still look like that. I feel like it is easy for misconceptions to arise by claiming facts from heuristically conceived visualizations like this. Perhaps a more reliable method would be to investigate the effective receptive field $R_{ij} = \| \partial z_j / \partial x_i \|$, to more quantitatively show that the latent space only encodes local information within n pixels of the corresponding latent coordinate. ------ 3.) Sec. 6: Again, I feel like the observations are not enough to support the conclusions. In Figs. 1 and 3, it is simply shown that the first 2 or 3 layers of a flow exhibit this behavior. Line 170 makes it sound like this is what happens even for an entire 10-layer flow, saying that OoD data and ID data are both treated in the same way. But Appendix Fig. 11 shows that the rear layers of the flow perform more complex, longer-range operations. By the same logic, one could argue that it should be impossible for standard feed-forward networks to perform image classification, because the first two layers of an AlexNet or VGG learn to extract edges; edges appear in all classes and aren't informative enough to perform the classification. The critical factor to explain the OoD behaviour might lie in the dominating contribution of these first few layers to the Jacobian, but this is not addressed quantitatively. Perhaps a useful fact to add: [38] find a very good correspondence between PNG compression and log-likelihood scores. PNG uses a rudimentary 'flow', by predicting each pixel value based on its upper left neighbours and only encoding the residual $z_i = x_i - t(x_{neighbour})$ (https://en.wikipedia.org/wiki/Portable_Network_Graphics#Filtering) This is similar to what you found in Figs 1 and 3 for the first few layers.

Correctness: I think the experiments are performed correctly, but in my eyes do not allow for the conclusions to be drawn from them.

Clarity: Language wise, the paper is very well written and easy to follow and understand.

Relation to Prior Work: Related work is cited and addressed, although I do think some some connections to prior work were overlooked, which have already been mentioned above.

Reproducibility: Yes

Additional Feedback: ==================== Update post-rebuttal ==================== I appreciate Figs 1 and 2, I feel they are a start to testing and checking the hypotheses in a proper way. The other reviewers convinced me in the discussion, that the observations alone may be of sufficient interest to the community, and I am raising my score accordingly. However, I am still very concerned that the paper does not reach the standards of scientific work. Normally, hypotheses should be made, and then experiments designed to test those hypotheses. But in the paper, no further experiments or skepticism is applied to the results. I feel at least it should be clearly said that what is written are untested hypotheses motivated by the observations, not proven facts. Again, to repeat some seeming contradictions that could have been skeptically discussed to better understand the problem, or perform further experiments based on this: - [35], [38] use the same type of models with the same training method (=same inductive bias), but show good OoD detection using a different OoD metric than raw log-likelihood. This contradicts hypotheses in l. 98 and l. 133 - Even a standard ResNet has recognizable spatial correspondence between input and features if it only has a few layers and uses 3x3 convolutions, by construction (concerning rebuttal l. 39). It also has rough spatial correspondence between deep features and input, as demonstrated e.g. by GradCAM (reubuttal l. 41, 42). So considerations like possible max. receptive field etc could be alternative explanations of the observations.


Review 4

Summary and Contributions: The paper focuses on the problem of likelihood assignments in normalizing flows NFs. In particular, that out-of-distribution (OOD) datasets can have higher likelihoods than instances from the training dataset. The authors conjecture that NFs learn the data via pixel correlations and present empirical evidence. The authors further present modifications to the architectures of the networks that improve the likelihood for OOD detection.

Strengths: The authors present empirical evidence for their hypothesis, that Normalizing Flows suffer because of the coupling layer co-adaptation. The relevance of the paper is very high, as modeling the likelihood of the distribution of the training set is very important as compliment to discriminator models to detect when the predictions make sense and when they are receiving OOD data that they can’t handle.

Weaknesses: As mentioned in the paper, the likelihoods produced by Normalizing Flows are known to be sensitive to the complexity of the input data. Indeed, near pixel reconstructions are easier on simpler datasets. And although the paper suggests some approaches to ameliorate this issue when training directly on the pixels, it does not offer a proof that explains why NFs fail to capture higher level structures. Other models from the probabilistic circuits family [1] also learn from pixel data and use the pixel correlations. Furthermore, they also learn using MLE, provide normalized likelihoods, yet manage to handle the OOD detection problem much better. This is evidence that MLE optimization is not necessarily the culprit, even when taking into account the normalization required. Therefore, it is still not totally clear from this paper, why NFs assign the likelihood mass they way they do to OOD instances. [1] Peharz, Robert, et al. "Random sum-product networks: A simple and effective approach to probabilistic deep learning." 35th Conference on Uncertainty in Artificial Intelligence, UAI 2019. 2019.

Correctness: Although there are experiments that aim to back up the presented hypothesis, they do not offer irrefutable proof on why the likelihood mass gets assigned as it is.

Clarity: the paper is well written and leads the reader to understand the concepts and ideas introduced.

Relation to Prior Work: The authors present a comprehensive connection to prior work, and the implications as remarked by the insights into prior work sub sub section.

Reproducibility: Yes

Additional Feedback: Do you believe the experiments on the st-networks capacity, are showing that the bottlenecks are affecting the total likelihood (Fig 4) negatively?. Is it fair to say that even when they help better distinguish the datasets mentioned, the total effect is that the likelihood mass is going down for the in-training-dataset? After reading the rebuttal, I see the value looking at the paper as insights about inductive biases and OOD detection for NFs.

[Author Response · NeurIPS 2020]

We thank all reviewers for their feedback! We are encouraged to see that the reviewers found our paper "clear, interesting, and novel" (R1, R2), "investigating an important issue" (R1, R3, R4), well-placed "into the related literature up to some very recent works" (R1), and "essential reading for anyone using RNVP / Glow-style architectures" (R2).

In the paper we study inductive biases of coupling layer based normalizing flows and show how they influence OOD detection — which types of data are assigned low and high likelihood. We identify mechanisms through which flows learn to improve likelihoods simultaneously on all structured images, even with semantics unrelated to training data. We believe that our analysis of flow's biases will be valuable for both DGM and OOD detection research communities.

**R1.** Thank you for your positive review! You are correct that our analysis is specific to coupling layer based normalizing flows. However, we believe that the intuition presented in the Sec. 4 of the paper can be translated to other models. We hope that our work will inspire similar investigations into the inductive biases of other DGMs.

**R2.** Thank you for your positive and detailed feedback! We agree that likelihood is not necessarily the right measure to detect OOD data, and we are not advocating for it. In this paper we study the question of how flows assign likelihood to data including OOD inputs. While typicality explains how in some particular cases OOD detection with likelihood fails (e.g. for high dimensional Gaussian), we argue that it does not by itself explain the likelihood assignment of more complex models like normalizing flows. We will clarify this point and add a discussion in the updated version.

**R3.** Thank you for your detailed feedback. While we agree with some of the points you raise, we respectfully disagree with your assessment. We believe that our empirical results would be valuable to the community and our narrative is consistent with the observations. We will clarify the limitations of our observations in the updated version of the text.

**1)** We agree that flows are likely learning *some* infor-
mation about the semantics, but it has little effect on
the likelihood. Our paper identifies mechanisms through
which flows can improve likelihood of the data regardless
of the semantics. We will clarify that we are not arguing
that flows do not learn semantics. Flows produce samples
that resemble the training data semantically; however,

Figure 1: $dz_i/dx$

Figure 2: $\log s$ by layer

how exactly they generate samples has to the best of our knowledge not yet been studied deeply, e.g., it is not clear to what extent flows memorize training data, how well they generalize and consequently whether they learn semantics.
**2)** The main purpose of Section 5 is to demonstrate that the latent representations learned by the flows can be interpreted and we can observe the edges of the original images in the latent space. We agree that on its own the presented results do not prove that flows do not learn the semantic information about the data. However, this observation is novel and at least partly contradicts the intuition that flows perform very complex high-dimensional transformation of the data; it provides motivation for the subsequent sections. Averaging dequantization allows us to denoise the latent representations to more clearly demonstrate that the latent representations contain the edges of the original inputs. We will clarify these points further in an updated version of the paper. • We performed the experiment you suggested and visualized the contributions of the different pixels to the Jacobian. For most latent coordinates the receptive field is limited to the neighbouring pixels (as in Fig. 1 left) but some coordinates are affected by longer range dependencies (Fig. 1 right). Thank you for the idea for this experiment, we will include a detailed description in the updated version of the paper.
**3)** Note that in Fig. 3 of the paper we are using a flow with two coupling layers, not the *first* two coupling layers of a deeper flow. We agree that the visualizations of the deeper layers of the flows are less interpretable than the first layers due to the added noise, but you can still see the edges of the original inputs (or the outputs of the previous coupling layer) in the deep layers of the flows in Fig. 11 (e.g. you can clearly see the shape of the face in the $s$-activations for Celeb-A). • To test your hypothesis regarding the dominating contribution of the first layers in the likelihood, we visualize the contributions of the different layers to Jacobian log-determinant in Fig. 2. On average the contribution is relatively uniform across different layers while the first few layers show higher variance of the predicted scale $s$.

**R4.** Thank you for your review. We respectfully disagree with your assessment and we hope that you will consider raising your score. We would like to further clarify several misunderstandings. The primary focus of our paper is to provide empirical evidence for why flows assign high likelihoods to OOD data (Sections 5–7), and not to provide remedies to this issue. While we propose several approaches that improve the OOD detection results, we view them as understanding experiments that help us analyze how different design choices influence likelihood assignment on in-distribution and OOD data. • We do *not* argue that the poor performance of the flows comes solely from their normalization, MLE or learning from pixels. Instead, we argue that the OOD performance of a method is primarily decided by its inductive biases, and we argue that the inductive biases of the flows (analyzed in Sections 5–7) are not aligned with OOD detection (see Section 4 and Appendix A). • $st$-network capacity experiment: the smaller bottleneck size results in lower likelihood for train data, however, in the context of OOD detection we are interested in *relative ranking* of in-distribution versus OOD likelihoods. As mentioned above, this is not presented as a proposed solution but rather as a part of the analysis of flow's inductive biases.

[Meta-Review · NeurIPS 2020]

After discussion, three of the four reviewers agree that the paper should be accepted. The meta-reviewer agrees; the paper provides valuable insights into learning of flows with coupling layers and into how the inductive biases affect out-of-distribution detection. However, some points need improving, please take the reviewers' comments into account when preparing the camera-ready version and implement all changes promised in the rebuttal. Moreover, please make it clear from the very beginning, e.g. in the title or abstract, that the focus is on a specific type of flows. One reviewer (Rev3) is critical about the general approach taken in the paper, asking for a more sceptical discussion, while not disputing that the observations are of interest to the community. The meta-reviewer thinks that the experiments are actually very well thought out and the results well explained. It is further clear from the write-up that the findings are mostly empirical, so that the usual caveats of experimental work apply. That said, some conclusions need rephrasing to avoid misunderstanding, as promised in the rebuttal.